# EGM Toolbox—Interface for Controlling ABB Robots in Simulink

**DOI:** 10.3390/s21227463

**Published:** 2021-11-10

**Authors:** Paweł Obal, Piotr Gierlak

**Affiliations:** Department of Applied Mechanics and Robotics, Faculty of Mechanical Engineering and Aeronautics, Rzeszow University of Technology, al. Powstańców Warszawy 12, 35-959 Rzeszów, Poland; pgierlak@prz.edu.pl

**Keywords:** industrial robot, robot control interface, position–force control

## Abstract

The development of industrial robotics requires the use of increasingly sophisticated control algorithms. In modern tasks posed by industry, it is not sufficient for the manipulator to move along a programmed path, reaching individual points with the greatest accuracy. There is a need for solutions that can allow detection and avoidance of obstacles appearing on the robot’s path and that can compensate the path for low-repetitive workpieces, adjust the strength of the impact of manipulator tools on the workpiece or enable safe cooperation of manipulators with people. To support this development, this work proposes an interface for controlling industrial robots in the Simulink environment. With its use, we can easily test our control algorithms using an external controller without the need to write an extensive program in the RAPID language. The robot controller’s task is to control the drives to achieve the set trajectory.

## 1. Introduction

Industrial robots are being used in increasingly complex production tasks that require the use of more sophisticated algorithms for controlling the movements of manipulators and sensory systems. These include machining processes such as deburring, edge rounding, grinding, or polishing [1,2,3]. These processes use force control systems that control the force exerted on the tool by contact with the workpiece. For example, in the processes of deburring and edge rounding, the force control system corrects the tool path in order to eliminate the influence of inaccuracies of the workpiece surface, which cannot be avoided in the case of cast elements [4]. In grinding and polishing processes, the downforce of the tool has a large impact on the quality of the surface finish and the thickness of the material removed. There are different ways to implement force control systems. For example, studies [5,6,7,8] have proposed tools mounted on a manipulator, which control the effector force independently of the movements of the manipulator itself. The tool has an additional mobile axis that moves the tool’s tip to obtain a specific downforce. A separate controller, independent of the robot controller, is responsible for controlling the force. Another approach is to control the manipulator’s motion trajectory depending on the force exerted on its working tip [9,10]. Leading manufacturers of industrial robots offer just such a force control solution for their manipulators. They use a 6-axis force and moment sensor usually mounted on the manipulator’s flange [11,12,13].

Our experience of using a force control system in ABB robots (ABB Ltd., Zürich, Switzerland) for robotic machining prompted us to work on its improvement. The factory add-on integrated force control (ABB Ltd., Zürich, Switzerland) allows us to program the trajectory of the robot based on two strategies [14]:FC SpeedChange—consists in reducing the linear velocity of the tool when the force acting on the tool in the direction tangent to the programmed path exceeds the set values;FC Pressure—where the set downforce of the tool against the surface of the workpiece is maintained.

This solution has limitations that make it impossible to use for machining workpieces with low geometrical repeatability, e.g., cast workpieces. The excesses that result from the casting process can come in a variety of shapes and sizes. Excessive geometrical variance makes it very difficult to achieve reproducible process results by selecting Force Control parameters and appropriate tools. Therefore, work was undertaken to find a better force control algorithm, which is presented in more detail in [15]. The experimental work was carried out on a stand equipped with a SCORBOT-ER 4pc robot (ESHED ROBOTEC LTD, Rosh Ha’ayin, Israel). However, this robot is not suitable for research on robotic machining in industrial conditions. It has low lifting capacity and stiffness, which prevents it from operating with a heavy spindle for machining hard metal alloys. Therefore, it was decided to build a stand equipped with an industrial robot with high capacity, which allows the testing of modern control algorithms for industrial processes at full scale.

By default, robot controllers do not allow full access to the control of the manipulator’s drives. The trajectory of the tool center point (TCP) or individual joints is generated based on a program written in the programming language of a given robot manufacturer. The program defines points on the path along which the manipulator is to move, the maximum velocity with which the path should be followed, the spatial accuracy within which the point should be reached, and some other parameters. On the basis of the determined trajectory, the control system only determines the signals for individual drives. The robot programmer cannot interfere with the algorithm for determining the trajectory and controlling the drives. To deal with this problem, we could build our own controller, using only mechanical units of industrial manipulators available on the market [16,17]. Based on the available documentation of the manipulator, it is possible to select appropriate motor controllers and power supply systems. Such an approach was described in [18,19], where for Estun robots (Estun Automation, Nanjing, China), a controller was built based on an industrial computer with a real-time system, which generates a trajectory for individual manipulator drives and sends it in real-time via EtherCAT to drivers powering drives. There are many ready-made solutions of universal drive controllers for multi-axis machines on the market, allowing the control of open kinematic chains. However, this is a very costly and labor-intensive approach. In addition, it carries a significant risk of damaging the mechanical unit as a result of the incorrect configuration of the drivers. In addition, the accuracy and repeatability of the manipulator control must be reliably determined. This requires proper research, preferably with a laser tracker, which is also a very expensive device.

Some robot manufacturers make it possible to set the trajectory from an external device in real-time. KUKA robots (KUKA AG, Augsburg, Germany) have the Robot Sensor Interface (RSI) [20] option, which can be used to correct the position of the robot or TCP axis based on signals from external sensors measuring the current position of the manipulator. Correction data can be sent via an I/O system or Ethernet using the User Datagram Protocol (UDP.) Weng in [21] used an RSI to correct the position of a manipulator while drilling and milling a workpiece made of aluminum. Much greater possibilities are provided by the KUKA Sunrise.OS system software (KUKA AG, Augsburg, Germany). It allows an external controller to fully control individual drives and control the manipulator’s TCP trajectory, and even control its compliance [3,22]. Unfortunately, this software is only available for the KUKA LBR iiwa series (KUKA AG, Augsburg, Germany). The largest robot in this series has a 14 kg load capacity. It is therefore not suitable for use in machining. An innovative solution, which is universal, is the uniVAL controller (Stäubli International AG, Pfäffikon, Switzerland) for Stäubli robots. It allows an external controller to move individual manipulator drives, control the TCP trajectory and program entire sequences of paths. An ordinary programmable logic controller (PLC) can be used as an external controller. The manufacturer provides an application programming interface (API) for PLC controllers from various manufacturers, including: Beckhoff (Beckhoff, Verl, Germany), Siemens (Siemens AG, Berlin, Germany), Rockwell (Rockwell Automation, Inc., Milwaukee, WI, USA). This solution was used in [23] to implement a proprietary force control system in order to improve the quality of the drilling process in CFRP material. The role of the external controller was performed by the Beckhoff C6930 industrial computer (Beckhoff, Verl, Germany), which performed the position–force control of the manipulator. Ultimately, due to the experience in working with ABB robots, it was decided to choose this company’s robot to construct a new test stand. The robot controller is equipped with the software add-on External Guided Motion (ABB Ltd., Zürich, Switzerland), EGM for short. As in the case of RSI for KUKA robots, this add-on allows us to correct the programmed path of the robot but also allows control of the manipulator’s trajectory from an external device [24].

We decided to use the application in the Simulink environment (The MathWorks, Inc., Natick, MA, USA) as a platform for an external controller. Article [22] presents the KUKA Sunrise Toolbox (KUKA AG, Augsburg, Germany) that allows control of robots with the KUKA Sunrise.OS system from the Matlab environment (The MathWorks, Inc., Natick, MA, USA). The creators extended this tool with the *Simulink-iiwa interface* that enables control of KUKA robots in Simulink [25]. Simulink makes it possible to easily simulate and test all control systems, e.g., manipulators. It has several ready-made tools, such as the Robotics Toolbox, which includes functions for determining the transformation of reference systems, solving simple and inverse kinematics and dynamics tasks, etc. Currently, there is no tool in Simulink that allows control of ABB robots via EGM. This paper presents a proprietary solution of an EGM interface in the form of an S-function block for the Simulink environment. Its task is to exchange information about the state of the manipulator and send the trajectory to be performed by the robot. In addition, it receives data from a six-axis force sensor, which is part of the factory Force Control option. The block is universal and can be implemented in any Simulink application. The main novelty of the work is that it will be possible to simulate the operation of control systems and conduct experiments on a real object within one environment.

Section 2 describes the construction of the robotic test stand and the equipment enabling its use for testing force control systems. Section 3 describes the structure and operation of the ABB EGM Toolbox interface. As an example of the interface operation, Section 4 presents the results of an experiment carried out on the operation of the interface based on a simple positional force controller.

## 2. Construction of the Robotic Station

The robotic station consists of an ABB IRB 2400 industrial robot (ABB Ltd., Zürich, Switzerland) equipped with the Force Control and EGM systems. Force Control is an add-on to ABB robots that allows us to program the robot’s trajectory based on the measured force exerted on the robot’s tool by the environment. The EGM add-on allows us to move the robot with an external device. The described case is a program run in the Simulink environment using the Desktop Real-Time toolbox, which provides a real-time kernel for models in Simulink. A schematic diagram of the station is shown in Figure 1.

### 2.1. External Guided Motion

The EGM addition to the IRC5 controller system allows the manipulator arm to be moved based on signals from an external device [24]. The EGM can operate in three modes:*Position Stream*—the current and planned position of the manipulator are sent to an external device;*Position Guidance*—the robot follows the trajectory from an external device in real-time;*Path Correction*—the robot corrects the programmed trajectory based on data from an external device.

Data can be transferred to the EGM via a digital and analog I/O interface or Ethernet User Datagram Protocol Unicast Communication (UdpUc) using Google Protocol Buffers (Protobuf) to encode the data. EGM enables feedback on the current state of the manipulator to an external device, but only by sending data via the UdpUc. According to the EGM documentation, we can read information about:the manipulator TCP position;manipulator TCP orientation in Euler angles and quaternions;the angular position of individual connectors of the manipulator;controller operation status;controller clock time;test signal values;force sensor signal values.

The manufacturer ensures that the frequency of information exchange by UdpUc is 250 Hz. Such communication is the best solution to control the manipulator from the level of a program running on a PC. Moreover, it does not require the use of additional physical I/O systems, intermediating in the information exchange. A diagram of EGM communication with the Simulink program as an external device is shown in Figure 2.

The exchanged data is encoded according to the Protobuf protocol. This forces the use of a serialization procedure for data sent to the EGM and deserialization of the received data so that it can be processed in Simulink. This protocol allows data to be transferred in a shorter time than standard protocols based, for example, on encoding information in ASCII. Unfortunately, Simulink does not have ready-made tools to conduct communication in accordance with the Protobuf protocol. Therefore, a custom S-function block was written to process the data and communicate with the EGM.

The EGM system is an intermediate arrangement between the external controller and the system generating control signals for individual manipulator drives. Through the EGM, it is possible to control the motion of the manipulator in joint space or in a task space defined in relation to the selected coordinate system. The EGM receives the set position and velocity data of the manipulator, then determines the set axis velocity from the formula:(1)speed=k·(pose−pose_ref)+speed_ref
where k—proportional gain factor, pose_ref—reference position, pose—set position, speed_ref—reference velocity. The set velocity reference value is transmitted to the motion control of the mechanical unit, which controls the individual drives.

### 2.2. Force Control

The Integrated Force Control add-on is not only a controller software extension package allowing the programming of paths in accordance with the chosen strategy. The add-on includes a 6-axis force/torque sensor along with a power system. The sensor can measure the force and torque acting on it as components defined in a Cartesian frame of reference $x_{S}y_{S}z_{S}$. The sensor can be mounted on the manipulator’s arm between the flange and the tool, as shown in Figure 3.

The force sensor can also be mounted onto a tool located off the robot’s arm, on the mating tool. In that case, the robot holds the workpiece and presses it against the tool in its working space.

## 3. Construction of the EGM Interface in Simulink

To test our own manipulator control algorithms, we used the Simulink environment with the Desktop Real-Time Toolbox, which allows the program to run in real-time. Support for communication with the EGM requires the use of Protobuf libraries to serialize and deserialize the information exchanged. The program of the S-function is written in C++. Once compiled, it can be implemented as a block in any Simulink application. The task of this block is:establishing a connection with the EGM;deserialization of the received data in accordance with Protobuf and transmitting them to the appropriate outputs from the block;serialization of the data transmitted to the block inputs according to Protobuf and sending them to the EGM.

During S-function tests, it was found that the EGM in *Position Guidance* mode can only receive force sensor signal values when the Force Control system is activated. Moreover, while Force Control is active, it is not possible to move the robot using the EGM. Therefore, when Force Control is on, EGM works only as in *Position Stream* mode.

However, the function of reading the signals from the force sensor is necessary in order to experiment with force control algorithms. The signals from the sensor can also be read thanks to the test signals. The test signals correspond to the values of selected operating parameters in the memory of the robot controller, such as position, velocity, or torque generated by the axis drives of the selected mechanical unit. The robot controller allows us to read test signals simultaneously on 12 channels. Table 1 presents the test signals for the force control system.

Signals in the range 201–206 correspond to the measured force and torque values relative to the sensor reference system, while signals in the range 207–212 are values transformed to a user-defined reference system. Most often, this is a system related to TCP in order to be able to operate on values that correspond to the influence of the environment on the tip of the tool. In Robotware (version 6.11), the option to read test signals via EGM has not yet been implemented. Therefore, the option of reading test signals via an additional UDP socket was added to the S-function. On the other hand, a semi-static task was created in the controller, which works independently of the EGM. A program was written that takes values from selected test signals and sends them via UDP to the indicated IP address. The program must define which signals are to be sent to the external device and in what order.

Eventually, the S-function block took the form shown in Figure 4. Depending on the EGM settings, the input of the block can be:
TCP position in relation to the selected reference frame as a vector ${\left[x_{TCP},\;y_{TCP},\;z_{TCP}\right]}^{T}$, expressed in (mm);TCP orientation in Euler angles as a vector ${\left[rx_{TCP},\;ry_{TCP},\;rz_{TCP}\right]}^{T}$, expressed in (°);TCP orientation in quaternions as a vector ${\left[q_{1},\;q_{2},\;q_{3},\;q_{4}\right]}^{T}$;position of the manipulator connectors as a vector ${\left[J_{1},\;J_{2},\;J_{3},\;J_{4},\;J_{5},\;J_{6}\right]}^{T}$, expressed in (°);velocity of the manipulator’s joints as a vector ${\left[{\dot{J}}_{1},\;{\dot{J}}_{2},\;{\dot{J}}_{3},\;{\dot{J}}_{4},\;{\dot{J}}_{5},\;{\dot{J}}_{6}\right]}^{T}$, expressed in (°/s);TCP velocity in xyz space as a vector ${\left[{\dot{x}}_{TCP},\;{\dot{y}}_{TCP},\;{\dot{z}}_{TCP}\right]}^{T}$, expressed in (mm/s).

However, at the output from the block, we get:
TCP position in relation to the selected reference frame as a vector ${\left[x_{TCP},\;y_{TCP},\;z_{TCP}\right]}^{T}$, expressed in (mm);TCP orientation in Euler angles as a vector ${\left[rx_{TCP},\;ry_{TCP},\;rz_{TCP}\right]}^{T}$, expressed in (°);TCP orientation in quaternions as a vector ${\left[q_{1},\;q_{2},\;q_{3},\;q_{4}\right]}^{T}$;position of the manipulator connectors as a vector ${\left[J_{1},\;J_{2},\;J_{3},\;J_{4},\;J_{5},\;J_{6}\right]}^{T}$, expressed in (°);values of forces and moments of the Force Control system, as a vector ${\left[F_{x},\;F_{y},\;F_{z},\;T_{x},\;T_{y},\;T_{z}\right]}^{T}$, expressed in (N) and (Nm);robot controller time, transmitted as a vector ${\left[hour,\;min,\;sec,\;ms\right]}^{T}$;number of the sample received from the controller;test signals, transmitted as a 12-element vector.

The first tests were carried out using the WireShark program (WireShark team, GNU GPL), which monitors data passing through an Ethernet network, and it was found that on average, the controller sends data from test signals every 4 ms. This frequency allows the data from the test signals to be used in parallel with the EGM.

The codes of the S-function program and the program on the robot controller that was used for testing were posted on the GitHub platform [26].

## 4. Experiment

In order to verify the operation of the S-function, a position–force control system was built in Simulink based on the algorithm described in [27]. The purpose of the algorithm is to implement the trajectory of motion in a direction tangent to the path and the trajectory of the force in the direction normal to the path [27,28]. To carry out the experiment, the stand shown in Figure 5 was used. The diagram of the selected control strategy is shown in (Figure 6b).

A ball caster (no. 1 in Figure 5) is used as a robot tool, which is pressed against the mating surface of the workpiece. The caster can move freely under pressure up to 700 N. As a workpiece, two metal flat bars with a thickness of 1 mm were used, spaced in the middle and fastened with clamps to the aluminum profile (no. 3 in Figure 5). Due to the high risk of damage to the manipulator when the tool hits the surface of the workpiece, the workpiece was attached to a pneumatic table (no. 2 in Figure 5) with an adjustable degree of compliance. Increasing the compliance of the mating surfaces prevents the tool from hitting the surface of the workpiece with excessive force. The compliance of the table-top can be adjusted by changing the pressure of the pneumatic cylinders that hold the table-top on both sides of the base (no. 2 in Figure 6a). It was assumed that as part of the experiment, the manipulator would make three runs along a straight line (no. 4 in Figure 6a) and press the tool against the surface of the workpiece with a constant force.

### 4.1. Controller Configuration

A *workobject* was defined in the robot controller memory, i.e., the user reference system $x_{O}y_{O}z_{O}$, relative to which the TCP position will be determined. The *tooldata* variable was also defined, containing information about the position of TCP ($x_{T}y_{T}z_{T}$) in relation to the manipulator flange as well as the weight of the tool and the location of its center of gravity. The EGM system was configured to run in *Position Guidance* mode with the option to control the TCP position (without orientation change) in the flat frame of reference $y_{O}z_{O}$ of the user. This configuration allows the manipulator to be controlled in the set plane, excluding motion control in the direction of the $x_{O}$ axis. The robot controller calculates the solution of the inverse and straightforward kinematics problem. The external controller receives data on the position of TCP in the task space, i.e., in the $x_{O}y_{O}z_{O}$ system. Figure 6a shows the individual frames of reference.

Configuration and commissioning of the EGM are performed in a program launched on the IRC5 robot controller. For the experiment, a program was written that set the manipulator TCP to the starting point of the planned path (point A in Figure 6a), set the EGM parameters, and waited for connection to an external device. At this point, the Simulink application running on the PC can establish a connection with the EGM and take control of the mechanical unit.

### 4.2. Control System Application in Simulink

In Simulink, a position–force control system was built, a diagram of which is shown in Figure 7. The trajectory of force $F_{nd}\left(t\right),\;{\dot{F}}_{nd}\left(t\right)$ and the trajectory of motion $c_{nd}\left(t\right),\;{\dot{c}}_{nd}\left(t\right)$ are transmitted to the input of the controller. The set positional-force trajectory used in the experiment is shown in Figure 8.

Dynamics equation of motion of the manipulator in the joint space takes the form described in [28]:(2)$$M\left(q\right)\ddot{q}+C\left(q,\dot{q}\right)\dot{q}+F\left(\dot{q}\right)+G\left(q\right)=u+J{\left(q\right)}^{T}\lambda$$
where $q\in R^{n}$—the vector of generalized coordinates, $M\left(q\right)\in R^{n\times n}$—the inertia matrix, $C\left(q,\dot{q}\right)\dot{q}\in R^{n}$—the vector of centrifugal and Coriolis forces (moments), $F\left(\dot{q}\right)\in R^{n}$—the viscous friction vector, $G\left(q\right)\in R^{n}$—the gravity vector, $u\in R^{n}$—the control input vector,$\;J\left(q\right)\in R^{m\times n}$—an analytical Jacobian matrix, $\lambda \in R^{m}$—an interaction force vector expressed in the task space, n—the number of degrees of freedom of the manipulator, m—a workspace (task space) dimension.

The analytical Jacobian matrix is determined from the equations of the manipulator’s kinematics:(3)$$J=\frac{\delta_{c}}{\delta_{q}}\;$$
where c—the vector of Cartesian coordinates. The kinematics of the manipulator in the Cartesian coordinates is described by the function:(4)$$c=k\left(q\right)\in R^{m}\;$$

The adopted control system is described by the equation:(5)$$U_{PD}=\left[\begin{array}{c} U_{c\tau }\\ U_{Fn} \end{array}\right]$$
where $U_{c\tau }$ is responsible for minimizing the motion lag error in the tangent plane, and $U_{Fn}$ for minimizing the force error in the normal direction.

These control elements are defined as PD control:(6)$$U_{c\tau }=K_{P}{\tilde{c}}_{\tau }+K_{V}{\dot{\tilde{c}}}_{\tau }$$
(7)$$U_{Fn}=K_{FP}{\tilde{F}}_{n}+K_{FV}{\dot{\tilde{F}}}_{n}$$
where $K_{P}$ and $K_{V}$ are successive matrices of proportional and differentiating gains of the position control system, while $K_{FP}$ and $K_{FV}$ are successive matrices of proportional and differentiating gains of the force control system. The error of the motion trajectory implementation in Equation (6) was written as:(8)$${\tilde{c}}_{\tau }=c_{\tau d}-c_{\tau }$$
where $c_{\tau d}$ is the set TCP position in a direction tangent to the surface of the workpiece,$\;c_{\tau }$ is the actual TCP position in a direction tangent to the surface of the workpiece. The user reference system was defined so that the $x_{O}y_{O}$ axes are tangent to the plane of the workpiece. Therefore:(9)$$c_{\tau }=\left[\begin{array}{c} x_{T}\\ y_{T} \end{array}\right]$$
where $x_{T}$ and $y_{T}$ are the coordinates specifying the position of the TCP in relation to the user’s system $x_{O}y_{O}z_{O}$. The error of the force trajectory implementation in Equation (7) was written as:(10)$${\tilde{F}}_{n}=F_{nd}-F_{n}$$
where $F_{nd}$ is the set downforce in the direction normal to the surface of the workpiece, $F_{n}$ is the downforce measured by the sensor in the direction normal to the surface of the workpiece.

### 4.3. Setting the Parameters of the Control System

It was assumed in the research that the robot tool would move along the workpiece, making three passes in a straight line, smoothly changing direction at the ends of the workpiece. The relationship describing the set TCP velocity was adopted as:(11)$${\dot{y}}_{Td}={\dot{y}}_{Td\;max}\left(\frac{1}{1+exp\left(-c_{v}\left(t-t_{ns}\right)\right)}-\frac{1}{1+exp\left(-c_{v}\left(t-t_{nk}\right)\right)}\right)$$
where ${\dot{y}}_{Td\;max}$ is the maximum TCP velocity, $c_{v}$ is the rate of rise and fall of velocity, $t_{ns}$, $t_{nk}$ define the time range during which the function reaches its maximum value, t∈[0, 100] s, n=1, 2, 3. The set velocity was composed of three successive runs of this relationship.

It was also assumed that the tool downforce should smoothly reach a certain value of $F_{nd\;max}$, which will be maintained throughout the motion. Therefore, the set value of the force is described, like the set velocity, by the relationship:(12)$$F_{nd}=F_{nd\;max}\left(\frac{1}{1+exp\left(-c_{F}\left(t-t_{Fs}\right)\right)}-\frac{1}{1+exp\left(-c_{D}\left(t-t_{Fk}\right)\right)}\right)$$
where $c_{F}$ is the rate of rise and fall of velocity, $t_{Fs}$, $t_{Fk}$ define the time range in which the function reaches its maximum value, t∈[0, 100] s. The time of this course is given in Table 2.

The controller realizing the set trajectory requires the gain parameters to be set in accordance with the adopted control system (5). The parameters of the PD controller are written as the gain matrices $K_{P}=diag\left(K_{Px},K_{Py}\right)$, $K_{V}=diag\left(K_{Vx},K_{Vy}\right)$ for the controller controlling the position and the parameters $K_{FP}$ and $K_{FV}$ for the controller controlling the force should be selected. They were selected experimentally on the basis of several test runs. The values of the applied parameters of the controller are given in Table 3.

The PD controller gain values for the $x_{O}$ axis motion were set to 0 because the EGM was configured to control TCP motion only on the other two axes. This allowed us to check how the EGM system copes with maintaining a constant value of this coordinate excluded from the coordinate motion without the participation of an external controller. The control signals from the controller in Simulink were connected to the CartesianSpeed input of the EGM IRC5 S-function block. The values from this input are sent to the EGM, which interprets them as speed_ref, according to formula (1). The k factor was set to 0 so that the TCP velocity would be generated based on the velocity control signal from the external controller.

### 4.4. Results of the Experiment

Figure 9 shows the realized TCP motion path, on which it is possible to observe how the coordinate $z_{T}$ changes reflecting the shape of the workpiece surface against which the tool is pressed. The motion in the tangential direction is divided into three phases:The first phase begins the motion from point A to point B. As the motion begins, the force control system begins to increase the downforce of the tool in the direction normal to the surface of the workpiece.The second phase begins after reaching point B, where the direction of motion changes. The tool from point B begins to move towards point A. The downforce is still maintained.The third phase begins when it reaches point A, where the robot again changes its direction of motion, stopping at point B. The downforce decreases smoothly, reaching 0 at point B.

The place where the flat bars spread apart is distinguished on the graph. It can be seen that the robot in this place lowered the height by about 1 mm in relation to the flat bar surface in order to be able to maintain a constant downforce, bracing the tool against the table surface.

During the second pass, it was observed that the tool, leaving the fault between the flat bars, does not return to the position $z_{T}=-0.6$ mm, but reaches the position $z_{T}=-0.8$ mm. This is due to the frictional forces between the surfaces of the piston and the cylinder of the pneumatic table actuators, which prevent the table from fully returning to its original position. As a result, the tool has to lower its position to achieve the set downforce. Figure 10 shows the realized positional trajectory in the tangential direction and the trajectory of the force in the normal direction. The contact force curve shows that the force value is kept at the set value. The successive decreasing and increasing pressure peaks occur at the fault of the workpiece surface. The control system reacts quite quickly to sudden changes in the contact surface.

No large overshoots or oscillations in the force value that could lead to tool damage were observed. The graphs in Figure 11 show the robot errors. The error ${\tilde{c}}_{x}$ shows that the tool is deviating from the start position relative to the *x*-axis excluded in the EGM. The deviation value is within the acceptable range. The greatest changes can be observed at the moment of changes in the direction of motion as short-term peaks not exceeding ±0.3 mm. Large and fast changes in the force error were recorded, which resulted from the shape of the workpiece surface. The sharp ending of the edge of the fault between the flat bars causes a sudden loss of contact at the beginning of the fault and a sudden collision with the edge at the end. During the periods of travel on an even surface of the workpiece, the force value is kept with an error below ±5 N.

Figure 12 shows how the control signals of the PD controller changed for motion in the directions tangent to the surface of the workpiece and for the downforce in the normal direction.

The performed tests have shown that the program for controlling the manipulator in the Simulink environment works correctly. The implemented control algorithm followed the set trajectory, and the lag errors were small. The conducted experiment was recorded on video and is available on the website [26].

## 5. Conclusions

This paper presents the construction of an interface for controlling the motion of ABB robots for an application in Simulink, using the factory add-on External Guided Motion. The Simulink environment is a highly convenient tool for simulating the operation of manipulator control algorithms. Furthermore, the EGM Toolbox interface allows us to additionally conduct experimental research on ABB robots as real objects, without the need to switch to another platform and write an additional program. Thanks to this solution, it is possible to simulate and verify manipulator control systems and process the collected data within one programming environment.

In future work, we plan to identify the manipulator model parameters for different EGM configurations so that different control systems can be simulated before checking their operation on a real object. The identification will allow for the adjustment of more advanced methods of manipulator motion control, determination of the stability of control systems, and the initial selection of control system coefficient settings. Ultimately, we plan to implement the position–force control system on a stand, taking into account the workpiece’s inaccuracies of geometric constraints during machining. The experiments, the results of which were published in [15], show that it can effectively minimize considerable deviations of the manipulator tool from the set path, caused by unforeseen deficiencies in the material of the processed workpiece, with the use of a force control system. Excessively large cavities may damage the robot’s tool as well as unnecessarily deepen the cavities. Trials so far have been carried out on a small scale. To confirm the effectiveness of the new force control strategy in industrial conditions, we plan to conduct a series of tests for a full-size industrial robot, performing machining of workpieces made of hard metal alloys using professional tools used in industrial plants. A built-in interface for operating the EGM system in the Simulink environment will allow for easy implementation of new control system solutions as well as accurate and reliable verification of the suitability of these solutions for industrial robots.

## Figures and Tables

**Figure 1 sensors-21-07463-f001:**
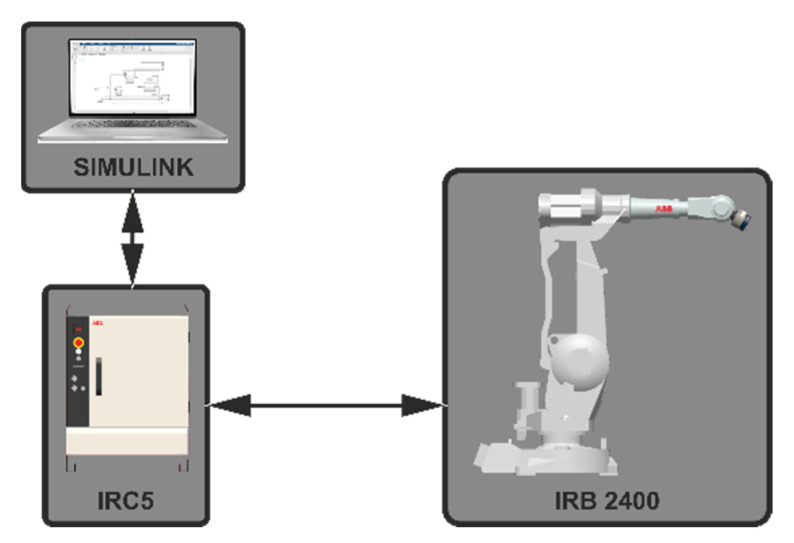
Diagram of the robotic station.

**Figure 2 sensors-21-07463-f002:**
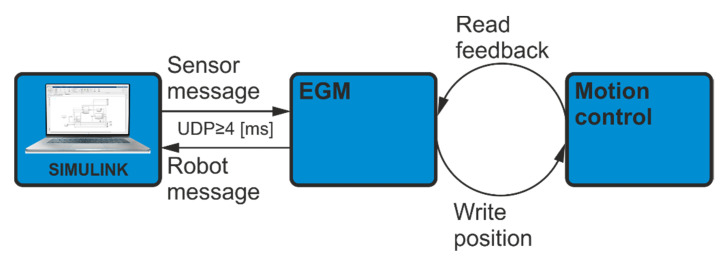
Communication diagram.

**Figure 3 sensors-21-07463-f003:**
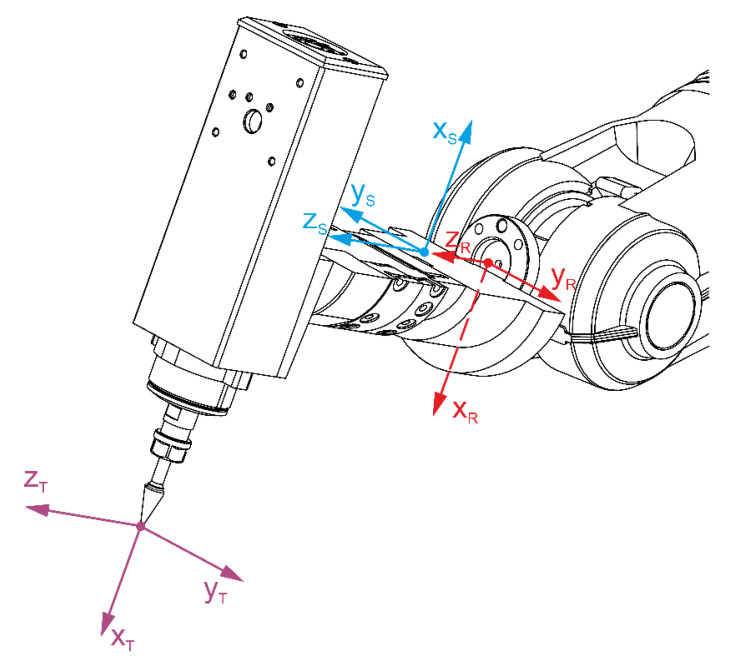
Diagram of the setting of a reference system in relation to the manipulator flange, $x_{R}y_{R}z_{R}$—reference system related to the manipulator flange, $x_{S}y_{S}z_{S\;}$ —reference system related to the force sensor, $x_{T}y_{T}z_{T}$ —reference system for TCP.

**Figure 4 sensors-21-07463-f004:**
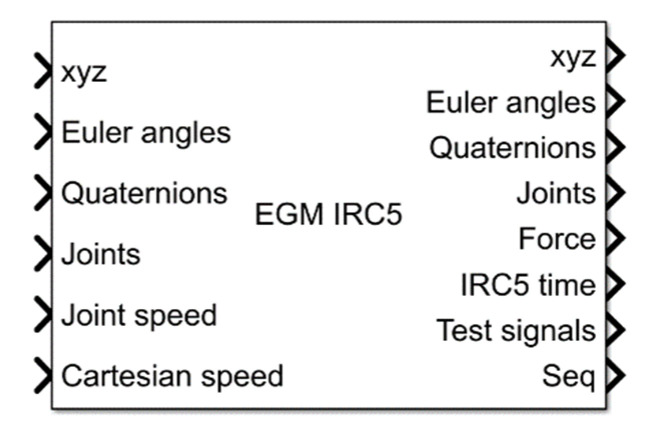
S-function block.

**Figure 5 sensors-21-07463-f005:**
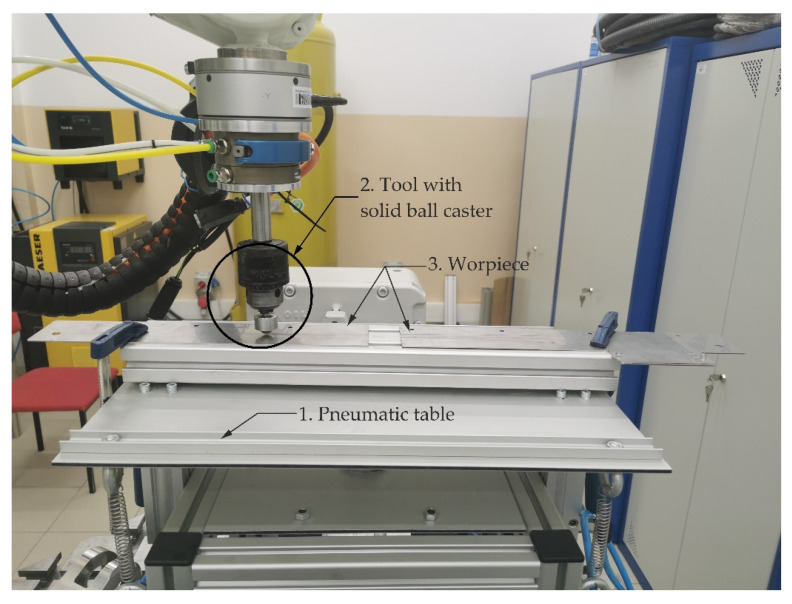
Photo of the test stand.

**Figure 6 sensors-21-07463-f006:**
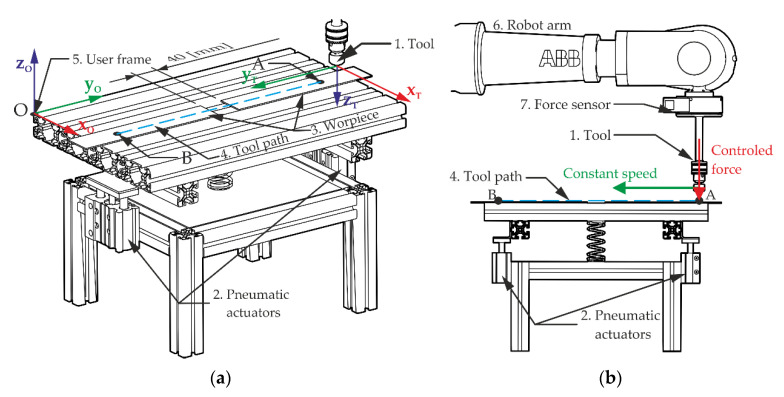
The scheme of the experiment: (**a**) Arrangement of coordinate systems on the test stand; (**b**) control strategy diagram.

**Figure 7 sensors-21-07463-f007:**
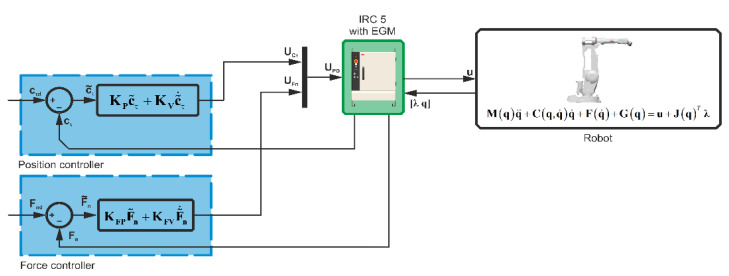
Diagram of the position–force control system for the ABB IRB 2400 robot.

**Figure 8 sensors-21-07463-f008:**
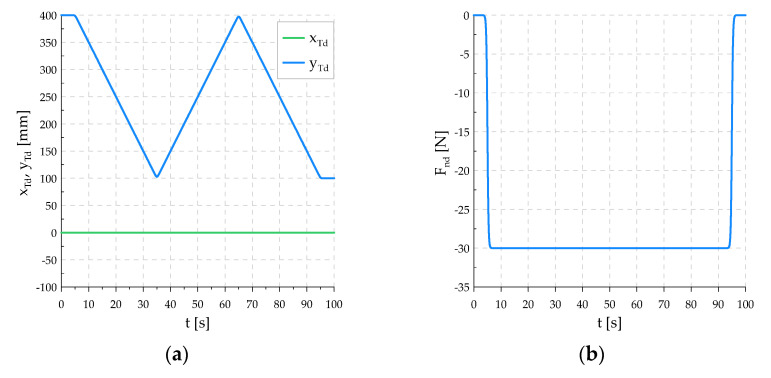
Graphs of set trajectory of control systems: (**a**) TCP coordinates on directions tangent to the surface of the workpiece; (**b**) downforce; (**c**) TCP velocity in tangent directions; (**d**) derivative of downforce.

**Figure 9 sensors-21-07463-f009:**
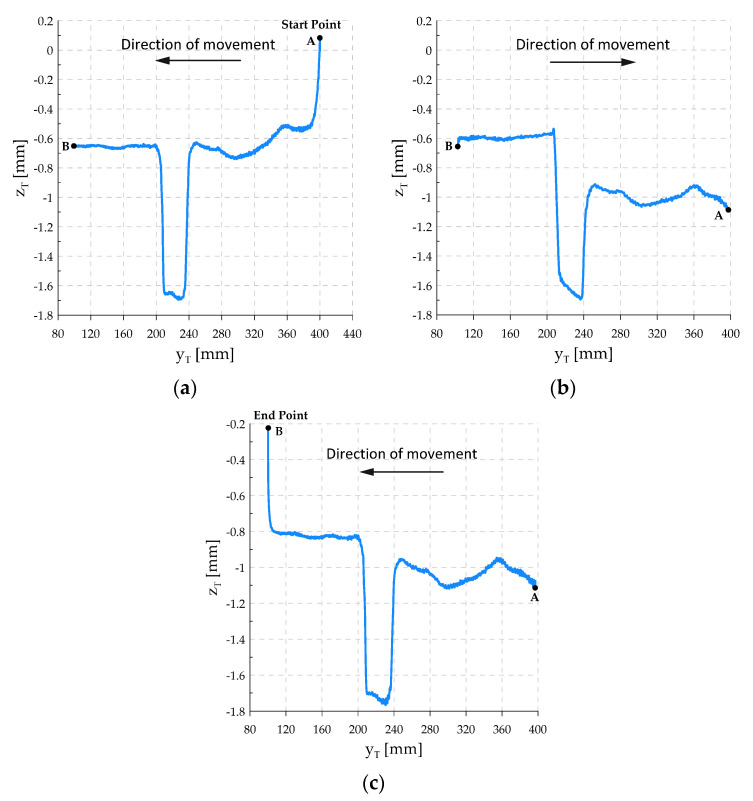
Graphs of the completed motion path, divided into 3 phases: (**a**) Motion phase No. 1 from point A to B, for the range $[t_{1s},t_{1k})$; (**b**) motion phase No. 2 from point B to A, for the range $[t_{2s},t_{2k})$; (**c**) motion phase No. 3 from point A to B, for the range $\left[t_{3s},\;t_{3k}\right]$.

**Figure 10 sensors-21-07463-f010:**
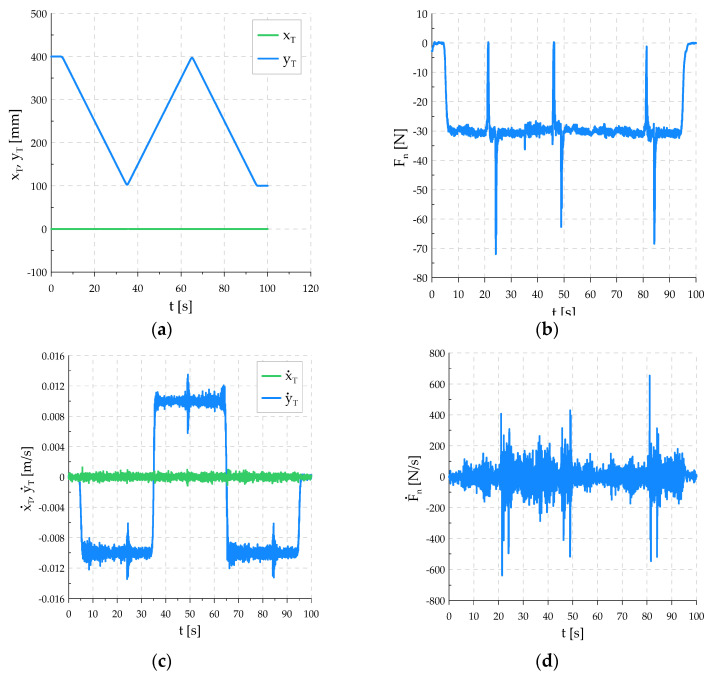
Realized trajectory of the system: (**a**) TCP coordinates on the directions tangent to the surface of the workpiece; (**b**) downforce; (**c**) TCP velocity in tangent directions; (**d**) derivative of downforce.

**Figure 11 sensors-21-07463-f011:**
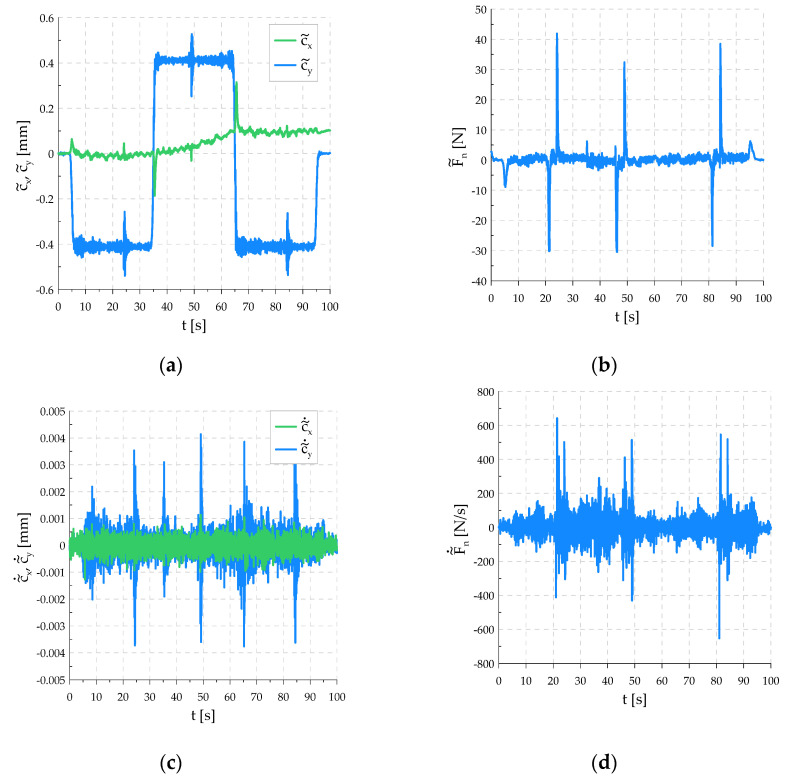
Lag errors: (**a**) TCP displacement errors in directions tangent to the surface of the workpiece; (**b**) downforce error; (**c**) TCP velocity errors in tangent directions; (**d**) derivative of downforce force error.

**Figure 12 sensors-21-07463-f012:**
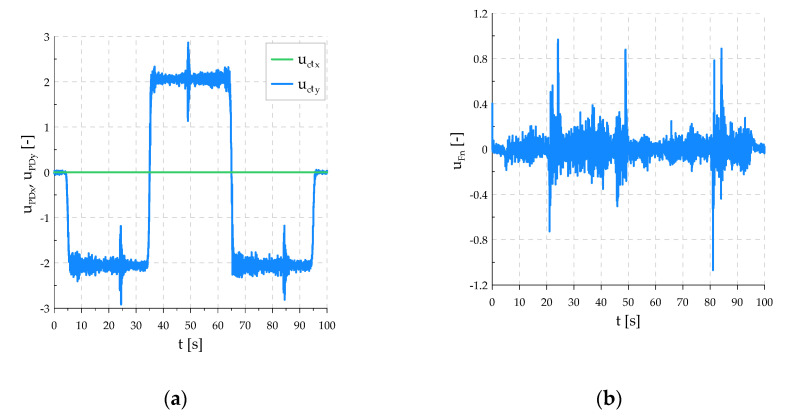
Control signals: (**a**) PD control signals in tangent directions; (**b**) PD control signal in normal direction.

**Table 1 sensors-21-07463-t001:** List of test signals for the force control system.

No.	Signal Number	Content
1.	201	Sensor frame, force with respect to the x-direction (N).
2.	202	Sensor frame, force with respect to the y-direction (N).
3.	203	Sensor frame, force with respect to the z-direction (N).
4.	204	Sensor frame, torque with respect to the x-direction (Nm).
5.	205	Sensor frame, torque with respect to the y-direction (Nm).
6.	206	Sensor frame, torque with respect to the z-direction (Nm).
7.	207	Force frame, force with respect to the x-direction (N).
8.	208	Force frame, force with respect to the y-direction (N).
9.	209	Force frame, force with respect to the z-direction (N).
10.	210	Force frame, torque with respect to the x-direction (Nm).
11.	211	Force frame, torque with respect to the y-direction (Nm).
12.	212	Force frame, torque with respect to the z-direction (Nm).

**Table 2 sensors-21-07463-t002:** Parameters of the set trajectory.

Parameter	Unit	Value
${\dot{y}}_{Td\;max}$	m/s	0.01
$F_{nd\;max}$	N	30
$[t_{1s},t_{1k})$	s	[5,35)
$[t_{2s},t_{2k})$	s	[35,65)
$\left[t_{3s},\;t_{3k}\right]$	s	[65, 95]
$\left[t_{Fs},\;t_{Fk}\right]$	s	[5, 95]

**Table 3 sensors-21-07463-t003:** Control system parameters.

Parameter	Value
$K_{Px}$	0
$K_{Vx}$	0
$K_{Py}$	5
$K_{Vy}$	0.1
$K_{FP}$	0.5
$K_{FV}$	0.05

## Data Availability

Data is contained within the article.

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
