# Peer review of "EGM Toolbox—Interface for Controlling ABB Robots in Simulink"

_sensors, 2021, doi:10.3390/s21227463_

Round 1
Reviewer 1 Report
The paper “EGM Toolbox – Interface for controlling ABB robots in Simulink” by PaweĹ‚ Obal , Piotr Gierlak proposed an interface for controlling industrial robots in the Simulink environment. The novelty of this paper is not clear. The following points should be considered:
- The contribution is not profound, also the title and content are so engineering. The topic is not so related with Sensors.
- Simulink is limited, which could restrict the use.
- There are two figures 8, which is an evitable mistake. Also, other figures are not well formatted.
- The contribution does not look enough.
Author Response
Sorry for the late reply. We sent a manuscript for the English editing services. Please see the attachment.
Kind Regards
Paweł Obal

Reviewer 2 Report
1. There are many phrasing and grammatical errors in the article, and the writing is not rigorous. It is recommended to modify it carefully. 2. The real-time simulation and control of robots have research significance, but it should focus more on the control algorithm of the robot, such as the timeliness of dynamics, rather than the way of engineering realization. Because all robot manufacturers provide offline and online control and simulation systems, and most of them support the open ROS.The control of robots combined with Simulink of Mathworks has been realized in many documents and technical specifications, such as the robotics toobox of Mathworks and the Robotstudio software of ABB. 3. Some errors stated in the text: 1) The abbreviated expressions "UdpUc" and "UPD"; 2) The expression of the unit [m], [°] should be (m), (°), etc. Please use parentheses in formulas and diagrams as much as possible, and there is no need to add parentheses in paragraphs; 3) The writing of vectors in the full text should be formatted, and there should be a separator "," between the vector elements, and it should be a column vector expression; 4) Do not write "," after the formula to avoid misunderstanding; 5) The writing format of the threshold interval in lines 336 and 343 should be a closed interval t∈[0,100], s; 6) The expression of the variables in Table 2 ⟨?2?, ?2?), <?3?, ?3?> is not uniform, 7) Please modify the layout of Table 2 and Table 3; 8) Line 348 ?? and ?? are incorrectly expressed. 4. The Lagrangian kinetic equation shown in Figure 6 is not described in detail. 5. The speed and displacement in Figure 8c and Figure 8a should be expressed in three constant speed stages.In addition, the three constant-speed processes shown in Figure 8c do not meet the control requirements of robotics, because sudden changes in speed (such as -0.01 directly to 0.01) will cause the quality of the processed surface to decrease. 6. Please explain the meaning of the unit N/s in Figure 8d. 7. It is recommended that Figure 8 and Figure 9 be taken seriously, especially for the analysis of each A->B or B->A process, such as the relative and absolute errors. 8. The 400*1mm aluminum plate was used in the experiment, if a thicker metal one or another material is used, will the same result be obtained?Author Response
Sorry for the late reply. We sent a manuscript for the English editing services. Please see the attachment.
Kind Regards
Paweł Obal

Reviewer 3 Report
The paper presents a novel control architecture for industrial robots using Simulink.
Although the research is sound and clear, there are a few aspects that need to be addressed.
The state of the art is well achieved, but it could be improved (please see “Control system of a medical parallel robot for transperineal prostate biopsy” or “Development of a Control System for an Innovative Parallel Robot Used in Prostate Biopsy”)
Regarding the S-function block description, how did you use quaternions?
Regarding the experiment description (lines 268-272), can you draw a sketch?
For the explanations within lines 275-285, can you show the indicated elements in the figure?
Lines 307-310: aren’t you repeating the information?
Fig. 8a (page 11) is hard to distinguish the two lines, please change the representation. Also, what are you trying to show with fig. 8d?
By the way, the figures numbers are wrong, you should check them.
Within fig. 7b, although the end-effector starts again from point B, there is an increase in the value of Z, why?
The information within lines 382-394 is hard to follow, maybe you can indicate on a sketch the phenomena.
The fig 8 (page 14) c and d have lots of noise, can you do something about it? Same for figs 9 and 10.
Can you define a metric that validates the newly developed control algorithm?
Author Response

(The authors gave the same response as above.)

Round 2
Reviewer 1 Report
It is not a professional correction type. The novelty and contribution are not enough.
Author Response
The novelty and the contribution of the article are presented in the penultimate paragraph of section 1. Moreover, we already explained it in the previous answer.
We don't know what the reviewer meant in the sentence: "It is not a professional correction type." Please explain your thoughts more clearly. Then we will try to take into account the comments.
Best regards
Paweł Obal
Reviewer 2 Report
The revised article is much better, wish you well.
Author Response
Thank you for accepting our amendments and appreciating our work.
Best regards
Paweł Obal
Reviewer 3 Report
The authors have addressed successfully all the issues.
The paper quality has definitely improved.
Author Response

(The authors gave the same response as above.)
